# Perspectives of Nursing Students towards Schizophrenia Stigma: A Qualitative Study Protocol

**DOI:** 10.3390/ijerph19159574

**Published:** 2022-08-04

**Authors:** Xi Chen, Jingjing Su, Daniel Thomas Bressington, Yan Li, Sau Fong Leung

**Affiliations:** 1School of Nursing, The Hong Kong Polytechnic University, Hong Kong, China; 2World Health Organization Collaborating Centre for Community Health Services, School of Nursing, The Hong Kong Polytechnic University, Hong Kong, China; 3College of Nursing & Midwifery, Charles Darwin University, Casuarina, NT 0810, Australia

**Keywords:** culture, schizophrenia, social stigma, nursing students, focus groups

## Abstract

Aim: This study aims to explore fourth-year nursing students’ knowledge of schizophrenia and their attitudes, empathy, and intentional behaviours towards people with schizophrenia. Design: This will be a descriptive qualitative study using focus-group interviews. Methods: Fourth-year nursing students on clinical placement in a hospital in Hunan province will be invited for focus-group interviews. Snowball and purposive sampling will be used to recruit nursing students for this study. Five focus-group interviews, each including six participants, will be conducted to explore participants’ knowledge, attitudes, intentional behaviours, and empathy towards schizophrenia. The interview will be conducted through the online Tencent video conference platform and the interview data will be collected through the same platform. All interviews will be recorded and transcribed verbatim and analysed with the approach of the content analysis supported by NVivo 12. Simultaneous data collection and analysis will be performed, and the interviews will be continued until data saturation is met. The findings of this study will be helpful in developing effective interventions to decrease the stigma toward schizophrenia among nursing students and those who study healthcare disciplines.

## 1. Introduction

Mental illness is a highly frequent disorder in which an individual’s cognition, emotion regulation, or behaviour is clinically significantly disturbed [1]. Mental illness stigma is defined as social stereotyping of, or discrimination against, people with mental illness [2]. A study involving 16 countries showed that the public had a high level of stigma towards people with mental illness [3]. Mental illness-related stigma is extremely damaging to people with mental illness, their families, and the health care system [4,5], and those with severe mental illness often face severe stigma [6]. Typical severe mental illnesses include schizophrenia, bipolar disorder, and major depression [7], of which schizophrenia is the most stigmatized mental illness [8,9,10].

Schizophrenia results in individuals’ significant impairments in perceiving reality and changes all aspects of their lives; it affects approximately 24 million people globally [11]. In China, an epidemiological study reported that the lifetime prevalence of schizophrenia among Chinese people was 0.6% between 2013 and 2015 [12]. A recent study reported that the age-standardized incidence rate of schizophrenia in China was 18.47/100,000 in 2019 [13]. The incidence rate implies that almost 8.46 million people in China had schizophrenia [14]. However, a huge number of people with schizophrenia were undetected by the healthcare system, as only 0.29 million (95% UI = 0.26 to 0.33 million) newly diagnosed schizophrenia cases were reported in 2017 in China [15]. Stigma is the main barrier to health-seeking behaviours, and the exploration of schizophrenia stigma is urgent.

People with schizophrenia face stigma in different aspects of their daily lives, such as in employment [16,17], friendships [18], romantic relationships [16,18], rehabilitation [19,20], social integration [20], and receiving medical treatment [21], which affects their life quality [10,22,23] and reduces their willingness to confide in others or ask for help [16,24,25,26,27]. Such stigmatisation can make them lose their self-esteem and ignore their own symptoms [24,28,29]. The loss of social status and familial respect caused by stigma [16] can even lead to suicide [30]. Meanwhile, stigmatising views held by health professionals can decrease their willingness to work in psychiatric departments [31,32], leading to a shortage of doctors and nurses [33].

Nurses constitute the largest occupational group in the healthcare sector globally [7]. Some nursing students in Mainland China also hold negative attitudes towards people with mental illness [34,35,36]. Compared with their USA peers, they were found to have poor knowledge of schizophrenia [37]. Nurses play a major role in the rehabilitation and treatment of people with schizophrenia [38]. Thus, exploring the perspectives towards people with schizophrenia among nursing students, as future nurses, will be invaluable in designing anti-stigma interventions for these students and ultimately improving the conditions of people with schizophrenia. Many quantitative studies have investigated nursing students’ knowledge of mental illness and/or their stigmatised attitudes and/or intentional behaviours towards people with mental illness [39,40,41,42,43]. Mental illness stigma is influenced by multiple factors [44], especially cultural factors [45]. But little quantitative research considers the cultural factors of stigma. Previous quantitative research often used some form of scale to assess the attitudes and intentional behaviours of nursing students’ stigma towards schizophrenia [46]. To develop specific and targeted interventions in decreasing nursing students’ stigma towards schizophrenia, it is necessary to understand their real perspectives. However, our knowledge of this area is lacking. Hence, a qualitative study will be more suitable for such an investigation. A qualitative descriptive research study generates data of three dimensions—who, what, and where—of events or experiences from a subjective perspective [47]. Philosophically, reality exists within a certain context, which is continuously changing and is perceived differently by different subjects. Thus, reality is characteristic of diversity and subjectivity, and it is highly recommended that the approach of constructionism and critical theories that use interpretative and naturalistic methods should be adopted in this research [48]. This study aims to obtain straightforward descriptions of nursing students’ experiences and perceptions towards schizophrenia. The qualitative descriptive approach is frequently used to provide straightforward descriptions of experiences and perceptions within a unique context [49]. In this type of research, the research process is inductive and dynamic, and the data collected from this phenomenon will not be over-transformed [49,50]. The advantages of the qualitative descriptive approach are both the questions’ subjective nature and participants’ diverse experiences, which can be recognized. Moreover, the outcomes will be reported straightforwardly, or in a way using similar terminology to the initial research question [51].

Some qualitative studies have explored the knowledge, attitudes, and behaviours of health professional students [52,53], nursing students [54,55], and medical students [56] towards mental illness. There is a lack of qualitative research focusing on schizophrenia stigma, even though schizophrenia is the most stigmatized mental illness [8,9,10], and the relationship between culture and stigma is given little attention in qualitative studies. Thus, further studies could be centred around nursing students’ perceptions of schizophrenia stigma. Unlike Western countries, China is a developing country with a culture of face-saving that is deeply influenced by Confucianism, Taoism, and Buddhism [57,58].

In view of the relationship between culture and the stigma, the development of anti-stigma interventions should be tailored to the Chinese culture. The findings of such a qualitative study may help to develop Chinese culture-specific interventions to decrease schizophrenia-related stigma among nursing students, as well as students of other healthcare disciplines and health professionals in China. 

## 2. Theoretical Framework

In 2006, Thornicroft put forward the idea that three domains have constructed the problem of stigma: “knowledge, leading to ignorance; attitudes, leading to prejudice; and behaviour, leading to discrimination” [59]. Although stigma has been conceptualized differently, attitudes and behaviours are two core elements in most conceptualizations [60]. Stigma can be recognized as a negative attitude towards people with mental illness, and public stigma consists of three base components: stereotypes, prejudice, and discrimination [61,62]. Empathy is recognized as how an individual perceives other people’s thinking and feeling, and figures out what makes someone give a response to other people’s suffering with sensitivity and care [63]. According to a meta-analysis report, empathy is a mediator of the relationship between intergroup contact and reduced prejudice [64]. A study reported that empathy is inversely related to stigma and can be used to predict stigma [65]. Some studies also reported that improving empathy could decrease negative attitudes towards people with mental illness or other specific populations, and function as a potential protective factor that can reduce stigma [65,66,67]. In different cultures, people have different understandings of stigma, treat it in different ways, and yield different results [68]. Culture and beliefs will affect people’s views towards mental illness [69]. “Notions of stigma are bound by culture” ([70] p. 1). Studies have shown that sociocultural and religious factors strongly influence stigmatising attitudes [71,72,73,74,75]. It is difficult to measure peoples’ real behaviours towards schizophrenia [76]. Thus, intentional behaviours will be measured instead. It is envisaged that understanding how nursing students’ knowledge, attitudes, intentional behaviours, empathy, and the cultural influence on them affect their perspectives of schizophrenia and support the future development of interventions to decrease schizophrenia-related stigma among nursing students.

## 3. Aims

This study will explore fourth-year nursing students’ knowledge of schizophrenia and their attitudes, empathy, and intentional behaviours towards people with schizophrenia.

## 4. Methods and Analysis

### 4.1. Study Design and Setting

This qualitative study will adopt a descriptive design focusing on fourth-year nursing students’ knowledge of schizophrenia and their attitudes, empathy, and intentional behaviours towards people with schizophrenia. The study will be conducted at a tertiary first-class hospital in mainland China that provides clinical training for more than 250 fourth-year nursing students from across the country each year. Data collection will be conducted through focus-group interviews based on an online interview platform (Tencent meeting, a very popular online interview platform in China).

It is recognized that focus-group interviews can help generate deeper and richer data in many scenarios, as they involve group interactions [77]. These data can help to construct the key components of the intervention [78]. Through focus-group interviews, data from the researcher and each participant can be collected, and new questions and answers can be generated through interactive verbal communication among the group members. Researchers can thus know their participants’ needs and feelings and explore the influence of cultural values and beliefs on them [79]. The consolidated criteria for reporting qualitative research [80] will be used to guide the reporting of the focus-group interviews and the writing of the qualitative protocol.

As mentioned above, it is widely recognized that cultural values influence the stigma surrounding mental illness. Focus groups involving in-depth interviews of a particular population group on a certain topic can be used to develop or modify relevant intervention protocols [77]. Thus, we chose focus-group interviews to collect the data from 25 July 2022 to 30 September 2022. Apart from the focus-group interviews, individual interviews will also be offered to any participant who does not feel comfortable discussing the sensitive topic of stigma in a group setting. 

### 4.2. Participants, and Recruitment

All the participants of this study will be fourth-year nursing students. WeChat or emails will be used for recruitment purposes. The principal investigator will be responsible for recruiting eligible participants and collecting their written informed consent. The inclusion and exclusion criteria are as follows: 

Inclusion Criteria:

Nursing students who (1) are on clinical placement at the hospital involved in this study; (2) are 18 years old or above; (3) can communicate in Mandarin; and (4) agree to participate in this study.

Exclusion Criteria:

Nursing students who (1) have no access to a computer, a smartphone, or any electronic device for joining the online interview. 

### 4.3. Sampling

“In focus-group research, the strategy is to use purposeful sampling whereby the researcher selects participants based on the purpose of the study [79] p. 452.” To gain rich information and to achieve maximum variation sampling from the participant, we will enrol fourth-year nursing students who satisfy the criteria and, if possible: (1) have contact experience with people with schizophrenia or mental illness, (2) are of male gender (most nursing students are female; thus, we want to enrol some male nursing students), (3) come from different provinces (it would be best to include some from ethnic minority groups), and (4) are very interested in studying mental health and willing to discuss their perspectives on our research topic. However, due to the COVID-19 pandemic and without contact information for the potential participants, it will be challenging and difficult to recruit a purposeful sample. Thus, the initial participant recruitment will adopt the snowball sampling method. Interested nursing students will be recruited through the affiliation of the principal investigator with a nursing college which can promote the study.

If possible, we will also ask the participants to help us recruit their peers who can meet the inclusion criteria of the aforementioned purposeful sampling. 

### 4.4. Sample Size

A sample size of 4–12 participants is recommended for focus-group interviews. However, as it will be challenging to manage more than 12 participants in online focus-group interviews, we will divide the participants into two or three independent focus groups [81], with six participants per group [79]. It is suggested that data collection should be controlled as data collection after data saturation leads to a waste of time and resources [82]; therefore, concurrent data analysis will be adopted. Data saturation determines the sample size; thus, an exact sample size cannot be determined before the focus-group interviews [83]. It is reported that the first 5–6 participants produce the majority of new information in the dataset, while little information is gained from the later participants, and 80–92% of early information is identified within the dataset from the first 10 participants [84]. Little new information will be collected when the sample size is close to 20 interviews [85]. Thus, we expect around 5 groups of focus-group interviews (about 30 participants) will achieve data saturation, (i.e., when no new codes or themes emerge). The focus-group interviews will be stopped upon reaching data saturation. 

### 4.5. Data Collection

An interview guide will be prepared for focus-group interviews based on a literature review and a research-group discussion. To make the interview questions clearer and easier to understand for nursing students, pilot interviews were used to collect suggestions about the interview questions, and then the interview guide was adjusted. The details of the questions included in the interview guide are provided in Table 1.

Tencent Video Conference will be used to conduct and record the online focus-group interviews. A sociodemographic questionnaire will be used to collect participants’ demographic information. The principal researcher, who has received postgraduate research training, will conduct the focus-group interviews. Meanwhile, one research assistant will use field notes to record the non-verbal expressions and emotional states of all of the participants. Since this will be an online focus-group interview, all participants are free to choose a comfortable place for themselves during the interview. When the interview starts, some warm-up questions will be raised first to build rapport between all participants. Each focus group interview will last approximately 1–1.5 h. All of the interviews will be recorded and transcribed. A research assistant and the principal researcher will check the transcripts’ accuracy by comparing the transcripts word-by-word with the recorded videos independently. They will compare the difference between the two versions and check the video recording together before agreeing on a final version. The final version of the transcripts will be shared with the participants to correct any discrepancies and provide additional clarification that might improve data accuracy. Non-verbal responses will also be recorded in the transcripts [86].

### 4.6. Data Analysis

Data analysis will be conducted after completing the data transcripts and the coding will be done as early as possible. The principal researcher will read the transcription many times to ensure his familiarity with the data, which will be helpful for the subsequent data analysis. The dimensions of analysis will rely on the interview data and focus on the specific objectives to explore participants’ knowledge, attitudes, empathy, and intentional behaviours towards people with schizophrenia.

#### 4.6.1. Descriptive Statistics

The demographic characteristics of the participants will be presented with descriptive statistics, including means, standard deviations, and percentages, where appropriate.

#### 4.6.2. Qualitative Content Analysis

Content analysis is widely used in qualitative research to discover the underlying meaning of words by quantification [87]. An inductive approach will be used for content analysis in this study, as it involves detailed readings of the raw data to derive concepts and themes, which allows findings to emerge directly from the analysis of the raw data, rather than from a priori expectations or models [88]. Furthermore, in qualitative content analysis, systematic coding is used to describe the meaning of the qualitative data [89]; the two most common approaches are manifest content and latent content analyses [90]. Manifest content analysis involves examining the surface structure of the text, while latent content analysis involves exploring hidden meanings of the text [91]. Both manifest content and latent content analyses have advantages in text interpretation, and a combination of both may ensure more consistent and accurate findings than either approach alone [91,92]. Thus, both types of content analysis will be adopted in this study to understand participants’ true views about schizophrenia. Simultaneous qualitative data collection and analysis have been reported to enhance the depth and quality of data analysis [93]. Thus, concurrent data analysis will be used in the interviews, and the interviews will be stopped upon reaching data saturation. 

The five steps of qualitative content analysis are described below: data preparing and organising, reading and memoing, data coding, generating categories, and presenting the description and themes [86].

NVivo 12, a qualitative data analysis software, will be used to manage the data. An experienced qualitative researcher will also be invited to code the transcripts independently, and the coding outcomes of the principal researcher and the qualitative researcher will be compared. If some disagreement occurs, the research team will discuss the themes and findings, and make modifications until a consensus is reached. An outline of this focus-group interview study is shown in Figure 1.

### 4.7. Issues of Data Trustworthiness

Qualitative validity refers to the extent to which the study findings are accurate at every step of the analysis, while qualitative reliability refers to the extent to which the study findings are consistent between different researchers and projects [93]. Credibility, dependability, transferability, and confirmability are the four criteria that determine the trustworthiness of a rigorous qualitative study [94]. Peer-debriefing, member-checking, and all interview data will be transcribed into Chinese by a research assistant independently. Different categories and subcategories will be established after all data are analysed, and the research findings will be translated into English. This process will help to decrease the loss of meaning during translation and thus improve the credibility of the findings [95]. A sound audit trail and analytical memos will be maintained, and details of each step of the study will be recorded to ensure the dependability of this study. The information on the demographic and clinical characteristics and study context will be provided in detail, and the transferability of the findings will be increased in this study [96]. The researcher will reflect upon his actions to determine whether he has provided any misleading cues to the participants. Meanwhile, a research assistant will examine the video recordings of the interviews to identify any instances where the researcher might have manipulated the participants’ responses. Peer-debriefing will be used to examine the data-analysis process. Two trained research assistants will examine the processes of discussion and analysis to improve the confirmability of the data [97].

### 4.8. Ethical Issues and Data Safety

Ethical approval has been obtained from the Research Ethics Committee of The Hong Kong Polytechnic University (HSEARS20220127002 on 22 February 2022) and the Research Ethics Committee of the Hospital (KE202203129 on 18 March 2022) to conduct this study. Written informed consent will be obtained from eligible participants before data collection. Before filling in the information sheet, the participants will be reminded of the voluntary nature of their participation in the study, and their right to withdraw from the study at any time without any consequences for their clinical rotation. All information of the participants will be kept confidential and destroyed three years after completing the study.

## 5. Discussion

Exploring nursing students’ perspectives of schizophrenia stigma is a significant research topic, due to the lack of relevant knowledge. The findings of this qualitative study can inform the development of a Chinese culture-specific intervention to decrease schizophrenia-related stigma in China. Focus-group dynamic interaction is one of the methods to deal with sensitive, potential, and unpredictable questions within a group interview setting [98], whereas individual interviews may gain profound and accurate information of sensitive issues from participants. Most of our participants will be interviewed in focus groups as Chinese students might feel shy to express their opinions over a sensitive topic and would feel more at ease when accompanied by peers [99]. However, given the face-saving tendencies of some Chinese people [58], we will also consider offering individual interviews for students who feel uncomfortable with a focus-group interview. To ensure the data trustworthiness of this study, particular attention will be paid to data coding and data analysis, as highlighted in the earlier sessions. All interviews will be strictly transcribed by the principal researcher and a research assistant into written form to avoid the loss of information. Agreement on the codes and themes will be ensured among the research team members. More importantly, we will invite our participants to examine the findings of this data analysis and determine whether these findings reflect their real perspectives or opinions.

## 6. Limitations and Strengths

Nursing students with strong stigmatised views towards people with schizophrenia may be unwilling to participate in this study due to perceptions related to social desirability. This could lead to bias in participant recruitment. In addition, only fourth-year nursing students (from different universities) will be included in the focus-group interviews in view of the students’ theoretical knowledge of mental health. Thus, our sample may not be representative of the whole nursing student population spanning all degree years. However, the fourth year is the final year of the nursing degree, when nursing students’ stigma towards people with schizophrenia may affect their choice of nursing specialisation and their care for such patients in the years to come. Thus, we believe that exclusively including fourth-year nursing students is a potential strength of this study, as its results may guide the development of specific interventions to decrease these students’ stigma towards schizophrenia and encourage more graduating nurses to work in mental health departments.

## 7. Conclusions

To the best of our knowledge, this study will be the first to explore the perceived effect of Chinese culture on the stigma towards schizophrenia among nursing students. The study findings will inform the development of culture-specific interventions to decrease this stigma in Chinese nursing students and the nursing workforce. The findings may also stimulate the development of such interventions for decreasing the stigma towards other mental illnesses among nursing students, as well as students of other healthcare disciplines and health professionals in China.

## Figures and Tables

**Figure 1 ijerph-19-09574-f001:**
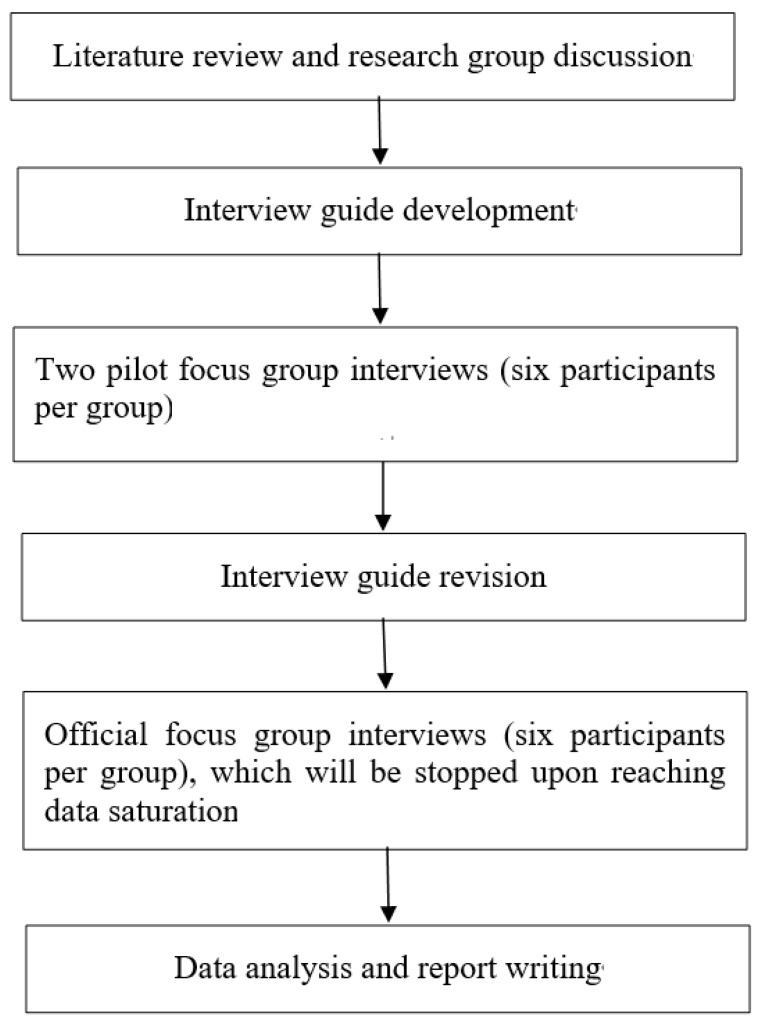
Outline of the focus-group interview study.

**Table 1 ijerph-19-09574-t001:** Interview guide of the focus groups.

1.	What is your understanding of schizophrenia (SZ)? (Probe: causal factors, manifestation, prognosis; where did you obtain such information?)
2.	What do you think about the life of people with SZ? (Probe: social support > intimate relationship > education > working)
3.	What are your experiences of interacting with people with SZ or mental illness? (If no personal encounter, any story you have heard?) [When > what happened > who > what did you say > How do you feel]?.
4.	How do you think Chinese people view schizophrenia compared to people from Western countries?
5.	What do you think of the views of traditional Chinese culture and religion on schizophrenia? [How do Confucianism, Taoism, and Buddhism view and deal with schizophrenia > According to the idea of traditional Chinese culture and religion, what are the causes of schizophrenia and how to deal with it?]
6.	How do you perceive the stigma often encountered by people with schizophrenia?
7.	What do you think are the main factors causing the stigma of schizophrenia?
8.	What do you think of caring for people with schizophrenia?
9.	From your view, how could an intervention program be used to decrease the stigma of schizophrenia?
10.	If you need to take part in a contact activity with people recovering from schizophrenia, what kind of activity will you recommend?
11.	What do you think of becoming a mental health nurse? (What are the motivations and barriers?)
12.	What are your suggestions to attract nursing students to work in the psychiatric department?

## Data Availability

This statement can be excluded as the study did not report any data.

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
