# Peer review of "Perspectives of Nursing Students towards Schizophrenia Stigma: A Qualitative Study Protocol"

_ijerph, 2022, doi:10.3390/ijerph19159574_

Round 1
Reviewer 1 Report
Thank you for the opportunity to review this revised manuscript. The authors have adequately addressed the concerns raised by my previous review.
Author Response
Thank you very much for your comments.
Reviewer 2 Report
The authors present the following abstract:
This study will explore fourth-year nursing students' knowledge of schizophrenia and attitudes, empathy, and intentional behavior toward people with schizophrenia Design: This will be a descriptive qualitative study using focus group interviews. Methods: Fourth-year nursing trainees at a hospital in Hunan Province will be invited to conduct focus group interviews. Four to five focus groups with six participants each will be conducted online on the Tencent videoconferencing platform. Simultaneous data collection and analysis will be conducted, and interviews will continue until data saturation is reached. Discussion: This study will be the first to use focus group interviews to explore nursing students' knowledge of attitudes, empathy, and intentional behavior toward people with schizophrenia and associations with Chinese culture. The findings will help develop effective interventions to decrease the stigma of schizophrenia among nursing students and those studying health disciplines.
The authors should significantly improve the abstract. When talking about the method in the abstract, talks about 4 or 5 groups. You should specify the exact number of focus groups and the dimensions you explored in the interview. Specify the type of interview and the procedure. How was the data analysis performed? By means of which program. Specify the main results. Eliminate the discussion dimension from the summary. The summary should not discuss the most salient results.
The objective of the study should be stated before the method, not before the theoretical framework, even if it is mentioned in the introduction.
You must specify specific objectives focused on the dimensions of analysis on which the interview was based.
You must specify the date of the resolution of the ethics committee.
At the end of the article, the authors must specify the tasks performed by each of the authors.
The English language should be revised. Incorrect expressions and errors continue to be detected.
Thanks to the authors for addressing the changes suggested by the reviewers.
Kind regards
Author Response
Thank you very much for your comments.

This manuscript is a resubmission of an earlier submission. The following is a list of the peer review reports and author responses from that submission.
Round 1
Reviewer 1 Report
Brief summary: This manuscript is the study protocol for a qualitative investigation of the stigmatizing attitudes Chinese nursing students may hold towards individuals with schizophrenia. The authors describe a plan to conduct 2-3 focus groups with 6 4th-year nursing students in each group using a video meeting platform. They plan to analyze focus group data using thematic analysis.
General comments:
1) The authors have provided a strong argument for studying Chinese nursing students’ attitudes towards schizophrenia and have postulated why stigma may be different for these students than in other populations studied. The topic is novel, and their work has the potential to inform future needed interventions to improve mental health care in China. While the review of literature and qualitative methods described in this protocol are sound, it is not clear that the study protocol on its own is offering a novel contribution to the field.
2) The use of focus groups instead of individual qualitative interviews bears some further discussion here. For discussion of a sensitive topic like stigma, is it possible that individuals might be more likely to share their honest beliefs if they are not in the company of peers?
Specific comments:
1) The exclusion of nursing students receiving treatment for a mental illness should be further elaborated; what type of impact are the authors anticipating mental illness might have on interview performance?
2) It is not clear exactly how many focus groups the researchers plan to conduct and how the two pilot focus groups will be incorporated into the study. While they state that they will stop data collection at the point of saturation, it might be helpful to describe the potential population (how many 4th year nursing students are in the total hospital population?) included to get an idea of how much data they anticipate collecting.
Reviewer 2 Report
Thank you for inviting me to review this work. First of all, I would like to thank and acknowledge the effort and work done by the authors of this study.
In order to follow and understand the comments made on your work, I inform you that I will respect the order and structure of your manuscript.
The theoretical framework of the study presented is insufficient to argue aspects of the objectives of this study.
The authors write the paper in the future tense. This is not correct if the study was carried out, it should be written in the past tense.
The attitudes, and the discriminatory processes suffered by people with mental problems, is widely studied and the authors establish a rigorous and deep analysis of this situation in their theoretical framework.
Measuring the effect of Chinese culture is an impossible aspect to measure, at least in this study; the research topics and their theoretical dimension are not adequately defined.
The authors neither measure nor assess the students' prior knowledge of schizophrenia, nor objectively determine their actual attitude.
In the student sample, the authors state in the inclusion criteria that the students were 18 years of age or older. If they were undergraduates, could they be 18 years old?
The methodology used and the procedure followed are not adequately described.
The results are not really discussed, the discussion section is poor and does not develop an argument based on the comparison of other studies with the findings of this study.
The conclusions are not real conclusions and do not respond to the objectives of the study.
Overall, the topic is of interest and the contributions are interesting, however, the authors should make the suggested changes and resubmit the paper. Please review the paper carefully to include the changes appropriately.
Please note that your paper requires improvements in the theoretical framework, mainly in the methodology section. The changes should be a substantial improvement to the paper.
The document must be reviewed by a native English speaker to correct grammar, expression, and vocabulary.
Please take the time to improve your papers.
Best regards
Reviewer 3 Report
The submitted manuscript aims to explore the knowledge of schizophrenia and attitudes, empathy and intentional behavior towards people with schizophrenia among fourth-year nursing students in mainland China, and the influence of Chinese culture on these variables.
To do this, it will use a qualitative methodology, consistent with the objective to be achieved.
The study is simple, not very ambitious and the introduction is well written, but the methodology is poorly developed.
It gives the impression to the reader that they are reading a teaching document and not a scientific article. The justification and importance of the development of qualitative methodology cannot be presented in a paper in the methods section.
They report that they will use an intentional sampling, not representative due to the difficulty of access to students, when for that reason a snowball sampling is much more accurate.
Regarding the procedure and collection of the data, it is again of general and diffuse writing, like a teaching manuscript. It does not refer to the selected categories, or how you have selected them.
In the analysis of the data he returns to the teaching and diffuse method, he does not speak of any theorist as a reference of analysis, I suggest that he read Taylor and Bodgan.
The results are presented as if it were a class work, more focused on a methodological paper than on the results of the study itself.
We do not know dates of studies
We don't know about starting or emerging categories
Speeches are not exposed
For all these reasons I reject the article, it has no scientific validity